# Complement factor H targeting antibody GT103 in refractory non-small cell lung cancer: a phase 1b dose escalation trial

Jeffrey M. Clarke[1,2] ✉, George R. Simon[3], Hirva Mamdani [4], Lin Gu[1,2], James E. Herndon II[1,2], Thomas E. Stinchcombe [1,2], Neal Ready[1,2], Jeffrey Crawford[1,2], Guru Sonpavde [4], Stephen Balevic[2], Andrew B. Nixon[1,2], Michael Campa [2,5], Elizabeth B. Gottlin [2,5], Huihua Li[2], Ruchi Saxena [2], You Wen He [2], Scott Antonia[1,2] & Edward F. Patz Jr [5] ✉

GT103 is a first-in-class, fully human, IgG3 monoclonal antibody targeting complement factor H that kills tumor cells and promotes anti-cancer immunity in preclinical models. We conducted a first-in-human phase 1b study dose escalation trial of GT103 in refractory non-small cell lung cancer to assess the safety of GT103 (NCT04314089). Dose escalation was performed using a "3 + 3" schema with primary objectives of determining safety, tolerability, PK profile and maximum tolerated dose (MTD) of GT103. Secondary objectives included describing objective response rate, progression-free survival and overall survival. Dose escalation cohorts included GT103 given intravenously at 0.3, 1, 3, 10, and 15 mg/kg every 3 weeks, and 10 mg/kg every 2 weeks. Thirty one patients were enrolled across 3 institutions. Two dose-limiting adverse events were reported: grade 3 acute kidney injury (0.3 mg/kg) and grade 2 colitis (1 mg/kg). No dose-limiting toxicities were noted at the highest dose levels and the MTD was not reached. No objective responses were seen. Stable disease occurred in 9 patients (29%) and the median overall survival was 25.7 weeks (95% confidence interval [CI], 19.1–30.6). Pharmacokinetic analysis confirmed an estimated half life of 6.5 days. The recommended phase 2 dose of GT103 was 10 mg/kg every 3 weeks, however further dose optimization is needed given the absence of an MTD. The study achieved its primary objective of demonstrating safety and tolerability of GT103 in refractory NSCLC.

Lung cancer continues to be a major public health issue and innovative therapeutic strategies are clearly needed to improve outcomes. The majority of patients treated for metastatic non small cell lung cancer (NSCLC) will develop disease progression within 1 year, with only a minority achieving long-lasting disease control[1,2]. There remains an urgent need for novel approaches to promote a durable response. A cancer target discovery program was designed to develop new candidate therapies based on an understanding of how tumors are eliminated by the native immune system. In syngeneic mouse models, IgG antibodies have been shown to be the initiating event in tumor rejection and are essential for ultimately stimulating adaptive immunity and an anti-tumor T cell response[3]. In our program we previously

[1]Duke Cancer Institute, Durham, NC, USA. [2]Duke University School of Medicine, Durham, NC, USA. [3]H Lee Moffitt Cancer Center—Advent Health Clinical Research Unit, Celebration, FL, USA. [4]Karmanos Cancer Institute, Wayne State University, Detroit, MI, USA. [5]Grid Therapeutics, Durham, NC, USA. ✉e-mail: jeffrey.clarke@duke.edu; edward.patz@gridthr.com

reported IgG autoantibodies targeting complement factor H (CFH) to be associated with early stage, non-metastatic NSCLC and increased time to recurrence, conceptually conferring a protective effect against cancer progression[4,5].

CFH is the dominant serum complement regulatory protein of the alternative pathway and influences multiple human diseases including cancer. CFH maintains homeostasis through inhibition of complement activation and the amplification loop that leads to cell lysis via two main mechanisms: acting as a cofactor for Factor I to cause inactivation of C3b and promoting dissociation of C3 convertase[6]. CFH is also implicated through mutation or overexpression in immune dysregulation (atypical hemolytic uremic syndrome, age-related macular degeneration), and pathogen-host immune evasion[7]. While CFH is predominately produced in the liver, multiple human cancer cell types express and bind CFH, resulting in resistance to complement-mediated lysis and promotion of local immunosuppression. In addition, high levels of CFH in the tumor cell membrane are associated with a poor prognosis[8–10].

Characterization of the anti-CFH autoantibodies in lung cancer patients found they were of a predominant IgG3 subclass[11]. Importantly, cancer patients with CFH autoantibodies demonstrate no obvious detrimental phenotype. As previously reported, high-affinity recombinant anti-CFH antibodies were produced directly from sorted, individual B cells against the specific CFH binding epitope from lung cancer patients who expressed CFH autoantibodies[12]. GT103 is a fully human-derived monoclonal IgG3 antibody that has anticancer activity in vitro and in vivo. Preclinical studies have shown that GT103 causes tumor-specific complement activation, antibody-dependent cellular phagocytosis (ADCP), and complement dependent cytotoxicity (CDC)[13]. Furthermore, in vivo, studies in syngeneic mouse models demonstrated no off-target effects, inhibition of tumor growth and metastasis[14] and most importantly, promotion of anti-tumor immunity through multiple mechanisms[12,15]. Most strikingly, GT103 treatment of tumor-bearing mice converts the tumor microenvironment (TME) from a pro-tumor state to an anti-tumor state, by decreasing immunosuppressive T regulatory cells (Treg), myeloid derived suppressor cells (MDSC), and pro-tumor N2 neutrophils, enhancing antigen-specific effector T cells, and activating B cells[13,15]. These changes occurred concomitantly with an increase in complement deposition in tumor tissues.

The basis for tumor specificity of GT103 is a presumed cryptic epitope in CFH, which is only exposed when the protein unfolds on the tumor cell surface. It is clear, by competition assays, that GT103 does not bind soluble serum CFH. It has also been shown by immunofluorescence that GT103 does not bind normal tissue CFH[13]. The crystal structure of a GT103 Fab fragment in complex with an epitope-containing peptide demonstrates a conformational change in the peptide relative to its structure in native CFH[12]. GT103 inhibits tumor growth of a wild-type tumor cell line but not tumor growth of a CFH knockout cell line[15]. This suggests that the antibody binding to the tumor cell-expressed CFH is required for its activity.

The epitope recognized by GT103 maps to a C3b binding site in CFH, leading us to hypothesize that the antibody blocks this site, allowing C3b to be deposited instead of destroyed, and thus promoting the activation of the alternative complement pathway[11]. However, in vitro, GT103 requires C1q but not factor B to activate complement lysis of tumor cells[13]. Therefore, GT103 appears to activate the classical pathway by first binding the cryptic epitope in CFH and then interacting with C1q via its Fc domain. In vivo, GT103 may bring about tumor cell killing by more than one mechanism. Given the close association between complement activation and restructuring of the immune program[16], our results suggest a linkage between GT103-mediated complement deposition in tumor tissue and the GT103-mediated changes in the TME observed in vivo[13,15].

In this phase I clinical trial, we present a dose escalation clinical trial of GT103 in patients with refractory, advanced stage NSCLC to assess safety and tolerability. Additionally, pharmacokinetic (PK) measurements and preliminary pharmacodynamic (PD) markers of GT103 are reported.

## Results

### Demographics and characteristics at enrollment

Between June 2020 and December 2023, 44 patients consented for the TOP1902 study across 3 institutions. Thirteen patients were deemed ineligible for various reasons (Fig. 1). Thirty-one patients ultimately received treatment with GT103. The median age of patients treated was 63 years (range 23–79), 61% were male, 74% were white, and 71% had Eastern Cooperative Oncology Group (ECOG) performance status 1. 61% of the patients had adenocarcinoma histology and 32% had prior history of brain metastasis at time of enrollment. The median number of prior lines of treatment was 3 (interquartile range [IQR] 2–4) (Table 1). PD-L1 tumor proportion scores were available for 25 patients, with 13 patients having PD-L1 scores >1%. PD-L1 status was not available for 6 patients. The most common mutations identified through standard of care genotyping included *P53* (*n* = 6), *KRAS* (*n* = 6), and *EGFR* (*n* = 5) (Supplementary Table 1).

### Safety

A primary objective of the trial was to determine tolerability and maximum tolerated dose (MTD). Only 2 dose-limiting adverse events (AEs) were noted during dose escalation: grade 3 acute kidney injury (0.3 mg/kg) and grade 2 colitis (1 mg/kg). Of note, the grade 3 acute kidney failure was deemed prerenal in etiology and related to grade 2 colitis in the same patient. The patient's renal function returned to normal following intravenous hydration. No dose-limiting toxicity (DLTs) were noted at the highest dose levels of 15 mg/kg every 3 weeks and 10 mg/kg every 2 weeks. No grade 4 or 5 AEs were reported during the trial (Supplementary Table 2). Treatment-related grade 3 AE occurred in 2 patients and included lymphocyte count decrease (3 mg/kg) and anemia (10 mg/kg Q2Wk). Common treatment-related AEs (≥10%) of any grade, included fatigue (*n* = 6, 19%), anemia (*n* = 5, 16%), diarrhea (*n* = 5, 16%), and nausea (*n* = 4, 13%) (Table 2). MTD was not reached during the trial.

### Treatment

The median number of cycles of GT103 received was 2.0 (IQR 2,4) and median duration of treatment 6.1 weeks (IQR 6.0, 12.3), (See Supplementary Table 3 for treatment duration by subgroup). The most common reason for stopping treatment was disease progression (*n* = 28, or 90%), followed by development of a new primary tumor, decline in performance status, and patient desire to withdraw from the study (*n* = 1 each). Secondary objectives included determination of objective response, progression-free survival (PFS), and overall survival (OS). The best treatment response was stable disease occurring in 9 patients; the disease control rate (DCR) was 29%. The median PFS was 6 weeks (95% CI, 6.0–6.1) with a 24-week PFS rate of 9.7% (95% CI, 2.5–22.9) (Table 3). Three patients maintained prolonged disease control at 26, 42 (censored) and 43.8 weeks (Figs. 2, 3). As of May 15th 2024, three patients were alive and survived for 42, 45, and 98.7 weeks, respectively. The median OS was 25.7 weeks (95% CI, 19.1–30.6). In an exploratory analysis by biomarker subclassification, *KRAS* positive patients had numerically longer PFS than KRAS-negative patients, median PFS 9.1 weeks (95% CI, 3.0–non-estimable) vs. 6.1 weeks (95% CI, 5.1–12.0) (Supplementary Tables 3, 4).

### PK

The PK of intravenous GT103 after single administration in adult patients with refractory, advanced-stage NSCLC was a primary objective. The PK profile was best characterized using a 3-compartment population PK model with multiplicative error and between-subject variability for central clearance (CL) and central volume of distribution (V). The time vs concentration profiles are noted in Supplementary

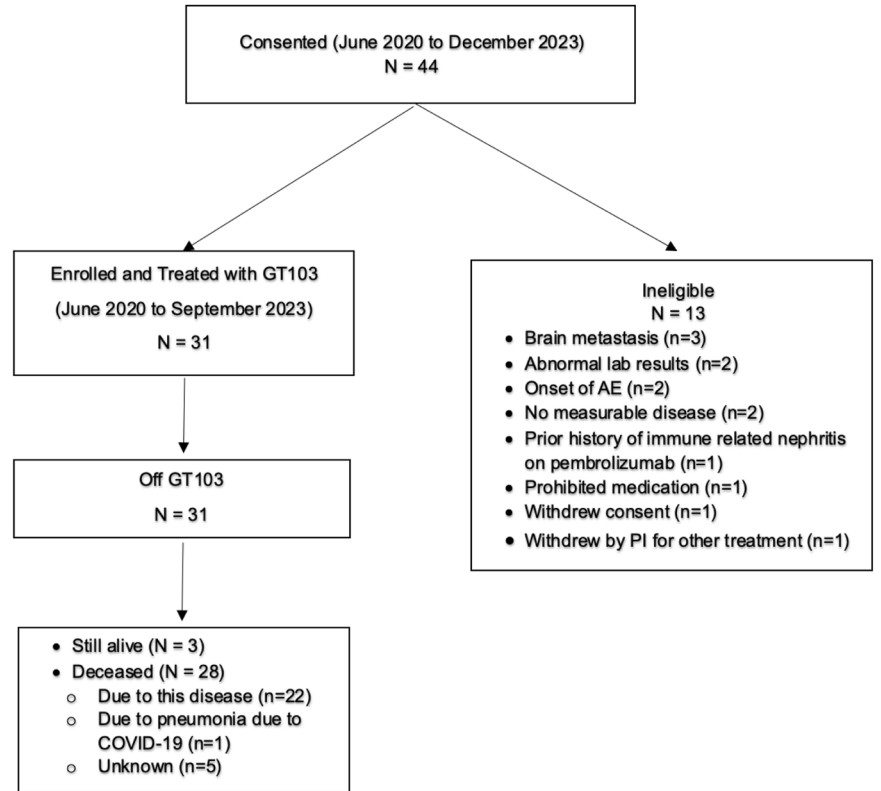

**Fig. 1 | CONSORT diagram.** As of May 15, 2024, 44 patients were consented and 31 patients were enrolled and received treatment on study.

Fig. 4. Exposures ($AUC_{0\text{-infinity}}$, $AUC_{0\text{-336h}}$, and $C_{max}$) were both considered dose proportional, with the beta-estimate being near 1 and the 90% CI not excluding 1. A summary of GT103 exposure is provided in Supplementary Table 5.

The typical population values were 16.8 mL/h for CL and 3786 mL for V, giving an estimated half-life of 156.2 h (6.51 days). However, a wide range of CL both within and across each dose level was observed, resulting in a large variability in post-hoc estimates of half-lives.

### Exploratory PD and immunohistochemical (IHC) correlative analysis

Serum levels of soluble C5b-9 (sC5b-9), the membrane attack complex that is a marker of complement activation, were measured in a pre-defined exploratory analysis as a candidate PD marker. Levels were measured at baseline (pre-dose), 24 h, day 8, and day 15. Change in sC5b-9 was expressed as % of pre-dose levels for patients at all dose levels. Modulation was seen by day 15 in a subset of patients during all dose levels following treatment with GT103 (Supplementary Fig. 2). A statistically significant association between baseline levels of sC5b-9 or day 15 change and PFS was not observed. CD55 and CFH were explored as predefined candidate IHC biomarkers and using archival tumor specimens from the time of initial diagnosis (Supplementary Fig. 3). Of note, higher levels of CD55 were associated with shorter PFS (hazard ratio 1.022, 95% CI 1.003–1.043, $p = 0.026$).

### Complement activity

Baseline complement activity was assessed as a post hoc analysis in the serum of patients at the two highest dose levels and was normal in all samples tested. This assured us that these patients had the capacity to activate complement.

## Discussion

In this first-in-human dose escalation phase 1b clinical trial, we describe the safety and tolerability as well as the PK and potential PD markers of GT103 in patients with advanced refractory NSCLC. The mechanism of action of GT103 is believed to be initiated by its binding of a cryptic epitope of CFH exposed on the tumor cell in the TME, causing tumor cell killing, and promoting an intratumoral immune response and limiting potential off-tumor effects. Indeed, GT103 was exceptionally well tolerated at all dose levels including the highest levels of 15 mg/kg every 3 weeks and 10 mg/kg every 2 weeks. The MTD was not reached. Based on the protocol, the recommended phase 2 dose of GT103 was 10 mg every 3 weeks. The safety and tolerability of the treatment are consistent with preclinical in vivo studies in which no off-target toxicities were seen, even at almost 1000x the highest dose given in this dose escalation cohort; this is consistent with the tumor specificity of GT103 and has implications for further drug development particularly in the neo-adjuvant setting. Given the mechanism of action, this type of immune-modulating therapy should be considered prior to other therapies that compromise the immune system's ability to generate a robust anti-tumor response. This includes surgery that removes the bulk of tumor antigens, and dissection of lymph nodes where cross-priming of T cell is essential for anti-tumor immunity.

Since this was a single-arm phase 1 trial without a control group, designed primarily for safety and tolerability, signals of efficacy need to be interpreted with caution. While no objective responses were observed from GT103 monotherapy according to the Response Evaluation Criteria In Solid Tumors (RECIST) criteria in this dose escalation trial, there are important caveats and indicators of potential clinical activity, including a DCR of 29%. The patient population in this study was heavily pretreated, with patients having prior progression on standard immune checkpoint therapy and multiple prior lines of treatment (median 3 lines). Several patients developed prolonged disease control beyond six months suggesting anticancer efficacy, potentially due to an adaptive immune response. Given the local inflammatory microenvironment created with GT103 as demonstrated in the preclinical studies, stable disease by RECIST nearing 30% of patients on this study, some with prolonged duration on trial, is a

**Table 1 | Demographics and disease characteristics of patients receiving GT103**

| | Total (N = 31) | Dose level | | | | | |
|---|---|---|---|---|---|---|---|
| | | 0.3 mg/kg Q3Wk (N = 6) | 1 mg/kg Q3Wk (N = 6) | 3 mg/kg Q3Wk (N = 3) | 10 mg/kg Q3Wk (N = 6) | 10 mg/kg Q2Wk (N = 5) | 15 mg/kg Q3Wk (N = 5) |
| **Age at enrollment (year)** | | | | | | | |
| Median (IQR) | 63 (56, 67) | 64 (56, 67) | 62 (58, 64) | 60 (60, 70) | 68 (63, 76) | 66 (56, 67) | 55 (54, 63) |
| Range | 23, 79 | 54, 69 | 57, 68 | 60, 70 | 51, 79 | 23, 72 | 45, 63 |
| **Sex, n (%)** | | | | | | | |
| Female | 12 (38.7) | 3 (50.0) | 3 (50.0) | 0 (0.0) | 3 (50.0) | 1 (20.0) | 2 (40.0) |
| Male | 19 (61.3) | 3 (50.0) | 3 (50.0) | 3 (100.0) | 3 (50.0) | 4 (80.0) | 3 (60.0) |
| **Race, n (%)** | | | | | | | |
| White | 23 (74.2) | 6 (100.0) | 4 (66.7) | 2 (66.7) | 4 (66.7) | 3 (60.0) | 4 (80.0) |
| Black or African American | 6 (19.4) | 0 (0.0) | 2 (33.3) | 1 (33.3) | 1 (16.7) | 2 (40.0) | 0 (0.0) |
| Asian | 2 (6.5) | 0 (0.0) | 0 (0.0) | 0 (0.0) | 1 (16.7) | 0 (0.0) | 1 (20.0) |
| **Performance Status (ECOG), n (%)** | | | | | | | |
| 0 | 9 (29.0) | 2 (33.3) | 1 (16.7) | 2 (66.7) | 2 (33.3) | 0 (0.0) | 2 (40.0) |
| 1 | 22 (71.0) | 4 (66.7) | 5 (83.3) | 1 (33.3) | 4 (66.7) | 5 (100.0) | 3 (60.0) |
| **Histology, n (%)** | | | | | | | |
| Adenocarcinoma | 19 (61.3) | 5 (83.3) | 4 (66.7) | 2 (66.7) | 3 (50.0) | 3 (60.0) | 2 (40.0) |
| Squamous cell carcinoma | 6 (19.4) | 1 (16.7) | 1 (16.7) | 1 (33.3) | 0 (0.0) | 2 (40.0) | 1 (20.0) |
| Non-small cell lung cancer NOS | 5 (16.1) | 0 (0.0) | 1 (16.7) | 0 (0.0) | 2 (33.3) | 0 (0.0) | 2 (40.0) |
| Poorly differentiated carcinoma | 1 (3.2) | 0 (0.0) | 0 (0.0) | 0 (0.0) | 1 (16.7) | 0 (0.0) | 0 (0.0) |
| **Disease Stage, n (%)** | | | | | | | |
| IIIA | 1 (3.2) | 0 (0.0) | 0 (0.0) | 0 (0.0) | 0 (0.0) | 0 (0.0) | 1 (20.0) |
| IV | 30 (96.8) | 6 (100.0) | 6 (100.0) | 3 (100.0) | 6 (100.0) | 5 (100.0) | 4 (80.0) |
| **Brain metastases at enrollment, n (%)** | | | | | | | |
| No | 21 (67.7) | 4 (66.7) | 3 (50.0) | 1 (33.3) | 5 (83.3) | 4 (80.0) | 4 (80.0) |
| Yes | 10 (32.3) | 2 (33.3) | 3 (50.0) | 2 (66.7) | 1 (16.7) | 1 (20.0) | 1 (20.0) |
| **Prior surgery for NSCLC, n (%)** | | | | | | | |
| No | 6 (19.4) | 0 (0.0) | 1 (16.7) | 1 (33.3) | 1 (16.7) | 2 (40.0) | 1 (20.0) |
| Yes | 25 (80.6) | 6 (100.0) | 5 (83.3) | 2 (66.7) | 5 (83.3) | 3 (60.0) | 4 (80.0) |
| **Prior radiation therapy for NSCLC, n (%)** | | | | | | | |
| No | 9 (29.0) | 3 (50.0) | 1 (16.7) | 0 (0.0) | 2 (33.3) | 1 (20.0) | 2 (40.0) |
| Yes | 22 (71.0) | 3 (50.0) | 5 (83.3) | 3 (100.0) | 4 (66.7) | 4 (80.0) | 3 (60.0) |
| **Prior radiation for metastatic brain lesion, n (%)** | | | | | | | |
| No | 21 (67.7) | 4 (66.7) | 3 (50.0) | 0 (0.0) | 5 (83.3) | 4 (80.0) | 5 (100.0) |
| Yes | 10 (32.3) | 2 (33.3) | 3 (50.0) | 3 (100.0) | 1 (16.7) | 1 (20.0) | 0 (0.0) |
| **No. of prior lines of systemic treatment** | | | | | | | |
| Median (IQR) | 3 (2, 4) | 3 (3, 3) | 3 (2, 4) | 4 (3, 10) | 3 (2, 3) | 2 (2, 3) | 4 (3, 6) |
| Range | 1, 10 | 3, 6 | 2, 7 | 3, 10 | 2, 5 | 1, 3 | 3, 9 |

promising result. In addition, 5 (16.7%) patients showed regression of measurable lesions though not meeting a threshold for response by RECIST. We recognize that in a phase 1 population, there is a possibility of selection bias toward slow-progressing tumors which may contribute to these findings. However, patients were progressing on prior therapy before they were enrolled and therefore any stability achieved during the trial could at least partially be attributed to treatment effect. Ultimately, an appropriate well-designed randomized trial in select patients will need to be performed to determine the true impact of GT103 on clinical outcomes.

The estimated population V of ~3–4 L is consistent with other monoclonal antibodies and suggests that the drug is limited to the vascular compartment (~5 L in adults[17]. Additionally, the estimated half-life of approximately 6.5 days for this IgG3 molecule is consistent with the reported half-life of IgG3 of ~7 days[18]. While there were some findings which could suggest non-linear drug disposition, both $C_{max}$ and AUC satisfied statistical requirements for dose proportionality. However, additional data may be necessary for definitive evaluation of linearity, and additional detailed PK analyses will be performed to optimize dose regimens.

While most immunotherapeutic approaches today commonly utilize IgG1 or IgG4 constructs[19,20], the IgG3 backbone is a unique feature of GT103 and may convey certain mechanistic advantages. GT103 was originally cloned as an IgG3 to recapitulate the subclass of anti-CFH autoantibodies associated with a favorable phenotype. Recent studies have suggested several fundamental advantages of IgG3 over other antibody subclasses. IgG3 fixes complement better than other subclasses, has improved effector function, and the longer hinge region permits improved binding of lower abundance targets[21,22].

Several biomarkers were explored in this study in an attempt to support the mechanism of action of GT103 and to eventually identify the patients who will respond to therapy. Exploratory analysis of

molecular characteristics of the patients' tumors suggests potential improved PFS with tumors with a KRAS mutation, however, these findings did not reach statistical significance. In addition, increased expression of CD55, another complement regulatory protein on the tumor cell surface was associated with potentially reduced survival, as suggested by prior studies[23,24]. Larger trials will be needed to identify tumor characteristics predicting benefit from GT103. Concentrations of sC5b-9 were measured in our trial at pre-treatment and during

treatment as a candidate circulating PD biomarker. Modulation was seen on treatment in a subset of patients receiving GT103 supporting its potential relevance. Although sC5b-9 has been measured in a number of various clinical conditions including infection, inflammation, trauma, and autoimmunity, little data are available on fluctuations with malignancy and anti-cancer treamtent[25]. Importantly, we report modulation of sC5b-9 in the setting of a complement-activating antibody in patients with cancer.

Additionally, complement activity was also assessed, as the ability to engage the complement system is essential for GT103's mechanism of action. While we could only evaluate 9 patients, complement appeared to be functional in all, unlike a prior study in chronic lymphocytic leukemia patients where 38% of patients tested had low complement activity, which could limit the efficacy of rituximab[26]. Future studies to assure complement activity prior to therapy in all patients will be important.

Ultimately, moving GT103 to an early line of therapy or combination immunotherapy may be needed to optimize clinical efficacy of GT103 in the metastatic setting. As described above, GT103 promotes multiple favorable effects on the TME in a murine model. Combination of anti-PD-L1 and GT103 treatment promoted additive tumor growth inhibition in vivo, while treatment of lung tumor cell lines with both antibodies increased CDC in vitro over treatment with one antibody alone[15]. Interestingly, other studies have demonstrated similar findings. Neutralizing antibody treatment against CD55 (decay accelerating factor) and CD59 results in complement and CD8 + T cell activation and inhibits tumor growth. Combination of antibodies against these complement regulatory proteins with anti-PD-1 results in synergistic tumor growth inhibition[27]. The safety profile of GT103 demonstrated in our phase 1 study suggests that the therapy would pair favorably with other immune checkpoint agents. Indeed, a phase 2 study of GT103 in combination with pembrolizumab is currently ongoing for patients with progression on standard frontline immunotherapy treatment in the 2nd/3rd line setting (NCT05617313). Further efficacy of dual therapy may also be seen in less treatment-refractory disease and earlier lines of therapy.

This study has limitations, particularly given the small number of patients. First, while all patients had advanced-stage disease, they each had multiple different lines of prior therapies that could have significantly altered their immune system and the clonal evolution of the tumor. Second, all tumor blocks available in this multi-institutional trial were obtained at the time of diagnosis and before any treatment. Expression levels can vary during therapy, and it remains uncertain if any of these markers may be predictive. Third, only patients in the two

**Table 2 | Treatment-related adverse events**

| Adverse event | Total (%) | No. of patients with maximum grade (%) | | |
|---|---|---|---|---|
| | | Grade 1 | Grade 2 | Grade 3 |
| Fatigue | 6 (19) | 6 (19) | 0 (0) | 0 (0) |
| Anemia | 5 (16) | 4 (13) | 0 (0) | 1 (3) |
| Diarrhea | 5 (16) | 5 (16) | 0 (0) | 0 (0) |
| Nausea | 4 (13) | 3 (10) | 1 (3) | 0 (0) |
| Alkaline phosphatase increased | 3 (10) | 3 (10) | 0 (0) | 0 (0) |
| Anorexia | 3 (10) | 2 (6) | 1 (3) | 0 (0) |
| Rash | 3 (10) | 2 (6) | 1 (3) | 0 (0) |
| Colitis | 2 (6) | 0 (0) | 2 (6) | 0 (0) |
| Infusion-related reaction | 2 (6) | 1 (3) | 1 (3) | 0 (0) |
| Lymphocyte count decreased | 2 (6) | 0 (0) | 1 (3) | 1 (3) |
| Platelet count decreased | 2 (6) | 2 (6) | 0 (0) | 0 (0) |
| Abdominal pain | 1 (3) | 1 (3) | 0 (0) | 0 (0) |
| Acute kidney injury | 1 (3) | 0 (0) | 0 (0) | 1 (3) |
| Alanine aminotransferase increased | 1 (3) | 1 (3) | 0 (0) | 0 (0) |
| Aspartate aminotransferase increased | 1 (3) | 1 (3) | 0 (0) | 0 (0) |
| Back pain | 1 (3) | 1 (3) | 0 (0) | 0 (0) |
| Creatinine increased | 1 (3) | 0 (0) | 1 (3) | 0 (0) |
| Dizziness | 1 (3) | 1 (3) | 0 (0) | 0 (0) |
| Headache | 1 (3) | 1 (3) | 0 (0) | 0 (0) |
| Hypotension | 1 (3) | 1 (3) | 0 (0) | 0 (0) |
| Increased pulmonary secretions | 1 (3) | 1 (3) | 0 (0) | 0 (0) |
| Nose bleed | 1 (3) | 0 (0) | 1 (3) | 0 (0) |
| Pruritus | 1 (3) | 1 (3) | 0 (0) | 0 (0) |
| Psoriatic flare-up | 1 (3) | 0 (0) | 1 (3) | 0 (0) |
| Stomach cramps | 1 (3) | 1 (3) | 0 (0) | 0 (0) |
| Vomiting | 1 (3) | 1 (3) | 0 (0) | 0 (0) |

**Table 3 | Summary of progression-free survival (PFS) and overall survival (OS), stratified by dose level**

| | Total (N = 31) | Dose level (mg/kg) | | | | | |
|---|---|---|---|---|---|---|---|
| | | 0.3 mg/kg Q3Wk (N = 6) | 1 mg/kg Q3Wk (N = 6) | 3 mg/kg Q3Wk (N = 3) | 10 mg/kg Q3Wk (N = 6) | 10 mg/kg Q2Wk (N = 5) | 15 mg/kg Q3Wk (N = 5) |
| **PFS** | | | | | | | |
| Events/N | 30/31 | 6/6 | 6/6 | 3/3 | 6/6 | 5/5 | 4/5 |
| Median Time[a] (95% CI) | 6.0 (6.0–6.1) | 6.6 (6.0–NE) | 6.0 (3.0–NE) | 5.4 (4.6–NE) | 5.9 (4.4–NE) | 12.3 (5.9–NE) | 6.0 (2.4–NE) |
| 24-week PFS rate (%) (95% CI) | 9.7% (2.5–22.9) | 16.7% (0.8–51.7) | 0.0% (NE–NE) | 0.0% (NE–NE) | 16.7% (0.8–51.7) | 0.0% (NE–NE) | 20.0% (0.8–58.2) |
| **OS** | | | | | | | |
| Events/N | 28/31 | 6/6 | 6/6 | 3/3 | 5/6 | 4/5 | 4/5 |
| Median Time[a] (95% CI) | 25.7 (19.1–30.6) | 29.1 (14.6–NE) | 16.5 (5.1–NE) | 20.4 (7.6–NE) | 30.2 (9.9–NE) | 30.6 (13.9–NE) | 19.1 (6.3–NE) |
| 24-week OS rate (%) (95% CI) | 51.6% (33.0–67.4) | 66.7% (19.5–90.4) | 16.7% (0.8–51.7) | 33.3% (0.9–77.4) | 66.7% (19.5–90.4) | 80.0 % (20.4–96.9) | 40.0% (5.2–75.3) |

[a]Weeks from initiation of GT-103.

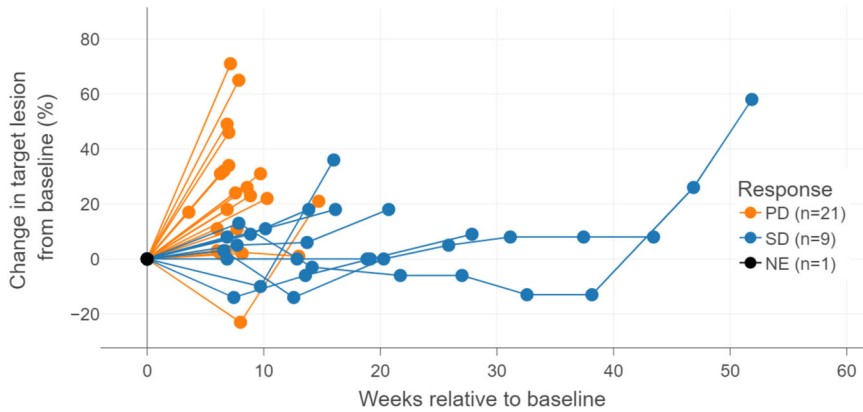

**Fig. 2 | Spider plot of individual subject tumor response.** Tumor size change as determined by RECIST criteria is plotted relative to baseline measurement.

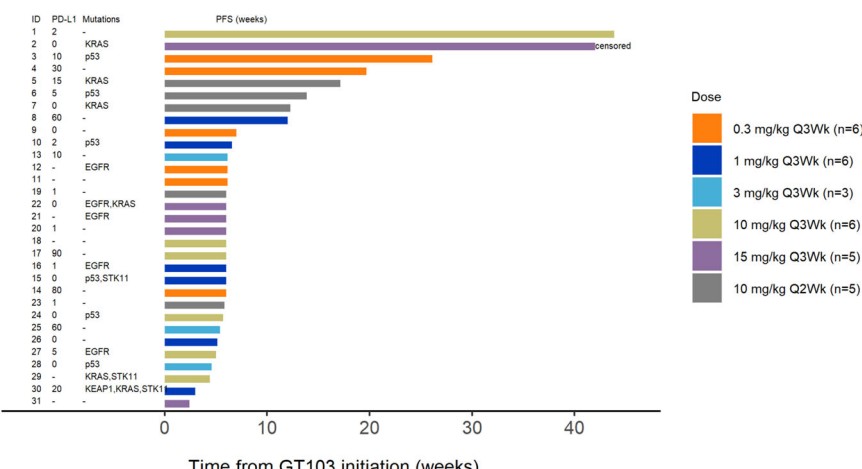

**Fig. 3 | Swimmer Plot showing individual patient progression-free survival (PFS).** Known tumor PDL1 score and genotype status are included for individual subjects in addition to PFS (weeks).

highest dose levels had serum samples available so that complement activity could only be determined in those select patients.

Regarding the population PK model, there are important limitations. First, no covariates were included in the dataset, and therefore, it is possible that the model could be further optimized after including patient or disease characteristics such as age, body weight, CFH concentrations, or tumor burden. Second, we did not evaluate non-linear PK models such as those that include target-mediated drug disposition or saturable pathways of elimination. Given the complexity of multicompartmental modeling, we estimated half-life assuming 1-compartment kinetics as an approximation. And lastly, we only evaluated the PK of GT103 after a single dosage. Additionally, it is important to note that noncompartmental analyses can produce biased results for biologic drugs, hence the reason for also performing the population PK analysis. Given this is a first-in-human dose escalation trial, and in the absence of an MTD, it is impossible at this time to select an optimal dosing level and schedule. Additional patients will be required to aid future modeling and dose optimization.

This study achieved its primary objective of determining safety: GT103 was found to be safe and very well tolerated. While overall survival was modest in this dose escalation study with an advanced-stage, heavily pre-treated patient population, the study achieved 50% survival at 6 months and there were several patients who attained prolonged disease control. Further development of GT103 particularly given its safety profile as an earlier line of therapy, in earlier stage disease as neoadjuvant therapy prior to surgical resection, and potentially in

patients with reduced expression of other complement regulatory proteins, will be able to elucidate its true potential as a treatment option.

## Methods
### Study design
The institutional review board of the three participating centers approved the study (Duke Cancer Institute, Karmanos Cancer Institute, and H Lee Moffitt Cancer Center—Advent Health Clinical Research Unit, Celebration, FL). The trial was conducted in accordance with Good Clinical Practice guidelines and the provisions of the Declaration of Helsinki. Patients were required to provide written informed consent before any study-related procedures. The study was registered with Clinicaltrials.gov on 03/16/2020 (NCT04314089).

TOP1902 was a first-in-human dose escalation trial of GT103 in advanced stage, refractory NSCLC to assess the safety of GT103 monotherapy (see Supplementary Information for protocol). Dose escalation was performed using a traditional "3 + 3" schema with GT103 dosed at 0.3 mg/kg for dose level 1. Treatment with GT103 was given IV every 3 weeks for 0.3, 1, 3, and 10 mg/kg dose levels. Based on safety and PK results, an amendment was implemented (November 2022) to incorporate two additional dose escalation levels at 15 mg/kg every 3 weeks and 10 mg/kg every 2 weeks. AEs as defined in the trial protocol were considered DLTs if they occurred during the first cycle of treatment and were deemed to be possibly, probably, or definitely related to study treatment as determined by the investigator. The protocol was opened at three institutions for patient enrollment.

Treatment was given until disease progression or unacceptable toxicity. Of note, the protocol was closed to enrollment prior to completing the 15 mg/kg every 3 weeks and 10 mg/kg every 2 weeks dose levels (5 patients treated at each level). This decision was jointly made by the sponsor and PI based on competing studies and projected accrual timelines, and not for toxicity reasons.

## Patients

Patients ≥ 18 years old with ECOG performance status 0 or 1, and adequate end-organ function, and histologically and/or cytologically confirmed advanced stage III, IV or recurrent NSCLC, and having disease progression on prior standard of care therapy were eligible to participate in the study. Subjects of both sexes (as self-reported) were enrolled sequentially, as were patients of any race. Patients had to have received immunotherapy (anti-PD-1/PD-L1) and a platinum-based chemotherapy either concomitantly or sequentially. Patients with *EGFR*, *ALK*, or *ROS1* alterations had to have received at least one prior tyrosine kinase inhibitor and prior chemotherapy (including at least one platinum doublet regimen). Key exclusion criteria included extracranial palliative radiation or anticancer therapies within 2 weeks from day 1 of study drug, intolerance to PD-1/PD-L1 axis drugs, or receiving chronic immunosuppression drugs other than prednisone <10 mg daily. Patients with leptomeningeal metastases or symptomatic brain metastases who continued to require glucocorticoids and/or antiseizure therapy were excluded. Patients were also excluded who had any severe and/or uncontrolled medical conditions, active infection, or previous history of cerebrovascular accidents, transient ischemic attack, myocardial infarction, pulmonary embolus or untreated deep vein thrombosis within 6 months. Patients with a history of interstitial pneumonitis of autoimmune etiology (including immune checkpoint-induced pneumonitis) which had been symptomatic and/or required treatment were not allowed. History of radiation pneumonitis was permitted if the patient had fully recovered and was off steroids at time of enrollment. The first patient and last patients were enrolled respectively on 6/17/2020 and 9/20/23.

## Assessments

Toxicity was assessed at every visit using the NCI Common Terminology Criteria for Adverse Events (NCI-CTCAE) version 5.0. Safety assessments were performed weekly during the first cycle of the dose escalation cohort, then every three weeks for subsequent cycles and every 2 weeks for the 10 mg/kg every 2 weeks dosing schedule. Safety assessments included vital signs, ECOG performance status, medical history, physical examination, review of concomitant medications, complete blood count with differential, and chemistries with liver function tests. After discontinuation of study treatment, subjects continued to have safety assessments for 30 days (±14 days) after the last dose of study drug.

Tumor response was assessed using RECIST version 1.1 and iRECIST by independent review. Baseline radiographic imaging included CT scan of chest/abdomen/pelvis with or without contrast and/or magnetic resonance imaging (MRI) of the abdomen/pelvis and all known or suspected sites of disease. Brain MRI was required for all subjects at baseline. Restaging scans of known sites of disease were performed every 6 weeks after the start of study treatment for the first 48 weeks and then every 12 weeks thereafter.

## Statistics

Descriptive statistics of median, IQR, and range for continuous variables, and frequency and percentage for categorical variables were provided to describe patients' demographics and disease characteristics at enrollment at each dose level. Median, IQR, and range were generated to summarize the number of GT103 treatments and total dose administered, as well as duration of treatment. Frequency and percentage were calculated to summarize best response to treatment,

reason(s) for going off treatment, number of patients with disease progression, and number of deaths. Treatment-related AEs were summarized. A spider plot was produced to visualize the tumor response and the tumor size changes relative to baseline. A swimmer plot was produced to display tumor growth characteristics over time.

The Kaplan-Meier method was used to estimate PFS and OS. DCR was defined as the proportion of patients who achieved a complete response (CR) plus those with a partial response (PR) plus those exhibiting stable disease (SD) following therapy. Given the unique mechanism of action of this immune modulating antibody, and unlike traditional chemotherapy, DCR was felt to be the most appropriate clinical metric to explore for efficacy, and not overall response rate. In preclinical models GT103 not only caused tumor cell death but promoted a robust immune response. The percent of viable tumor cells is reduced, and the percent of immune infiltrating cells is increased, which does not cause a significant change in size but produces a pathologic response. Thus, tumor size by non-invasive imaging may not necessarily change. In addition, early change in tumor size does not correlate with OS in advanced-stage NSCLC[28]. PFS was defined as the time from initiation of GT103 to disease progression or death from any cause, whichever came first. If a patient remained alive without disease progression at the time of analysis, PFS was censored at the time of last follow-up. OS was defined as the time between initiation of GT103 to all-cause death. OS was censored at the time of last follow-up if the patient was alive at the time of analysis. Median survival time and 24-week survival rate were estimated with 95% CI. Kaplan-Meier curves were generated to graphically depict PFS and OS for each dose level.

For sC5b-9, a univariable Cox proportional hazard model was used to assess the association between baseline levels of plasma sC5b-9 or day 15 change and PFS. An unadjusted Cox model was used to assess the association between IHC tests and PFS.

## PK analysis

Plasma for PK was collected immediately pre-dose, and at 2, 4, 8, 24 h (Day 1), and on Day 8, Day 15 after the first dose of GT103 on Cycle 1. PK samples were also collected immediately pre-dose Cycles 2–5. Plasma was isolated and stored frozen at −80 °C, then thawed and stored on ice for the assay. Quantitation of GT103 in patient plasma was carried out using an indirect ELISA. Briefly, biotin (B4639, Sigma Aldrich) and GT103 epitope peptide, GPPPPIDNGDITSFP(GGGK-biotin) (GenScript, Piscataway, NJ), were each diluted to 2 µg/ml in Dulbecco's Phosphate Buffered Saline with 0.1% (v/v) Tween 20 (DPBST). NeutrAvidin-coated 96-well plates (catalog #15129, ThermoFisher) were then pre-washed 4 times with DPBST. One half of the plate was incubated with peptide solution (100 µl/well) and the other half was incubated with biotin solution (100 µl/well). Plates were sealed and incubated while gyrating at 120 RPM for 30 min at room temperature using a Labline orbital shaker. Following incubation, plates were washed 4 times with DPBST. Standards were prepared by diluting human GT103 antibody (Catalent Pharma Solutions, Madison, WI, lot #D2-3776-02-01, 03OCT2018) into DPBST. Patient samples were appropriately diluted with DPBST and 100 µl/well of each sample and standard was loaded in duplicate on the peptide portion of the plate and on the biotin portion of the plate. Samples and standards were incubated gyrating for 90 min at room temperature. After washing the plate, goat anti-human IgG gamma-chain-horseradish peroxidase (HRP; Chemicon, catalog #AP504P) was diluted 1:8000 in DPBST and 100 µl added to each well. The plates were sealed and incubated as above for 60 min. After the final wash, 100 µl 1-Step ABTS (catalog #37615, Thermo Scientific) was added to each well, incubated for 30 min, followed by absorbance measurement at 405 nm. After subtracting the mean absorbance of the biotin wells from that of the peptide wells, the levels of GT103 present at each timepoint for each patient were determined from the standard curve.

For the calculation of PK parameters, the actual GT103 blood concentrations were derived by subtracting baseline antibody

concentration from the concentration post-dose. The first concentration beneath the quantifiable limit (BQL) was imputed to 1/2 the lower limit of quantitation (0.061 ng/mL) and estimated as if all the values are real, and all subsequent BLQ data set to 0. The GT103 PK assay was performed and analyzed in two batches: the first before the protocol amendment and the second following the protocol amendment.

PK data were analyzed with Phoenix NLME (Certara, Princeton, NJ, USA, v 8.4) using a population PK approach. The first-order conditional estimation with extended least squares or Quasi-Random Parametric Expectation Maximization algorithm was used to estimate model parameters. One, two, and three-compartment PK models for GT103 were explored assuming linear PK. Multiplicative, additive, and combined (additive plus multiplicative) residual error models were explored. Between-subject variability for PK parameters was estimated using an exponential function.

To derive GT103 exposures, a noncompartmental analysis was performed in Phoenix WinNonlin (version 8.4). The area under the curve (AUC) was calculated using the Linear-Up, Log-Down trapezoidal method. To assess linearity, exposures (e.g., AUC and maximum concentration [$C_{max}$]) were dose-adjusted. Additionally, the assessment of linearity was determined visually from plots by comparing exposure and dosage. A statistical analysis was conducted to investigate the dose proportionality using the power model, calculated as ln(parameter) = ln(dose)* β + intercept. For $C_{max}$ and AUC, the point estimate of β and its 90% CI was provided; a β estimate approximating 1 and a 90% CI containing 1 were considered dose proportional. Pairwise comparisons of doses were performed within and between each dose level and batch.

### sC5b-9 analysis in plasma

Blood was collected in EDTA tubes for isolation of plasma. The plasma was isolated, aliquoted, and frozen at −80 °C. For the sC5b-9 assay, samples were thawed and placed on ice. Concentrations of complement sC5b-9, also known as terminal complement complex (TCC), were assessed in plasma using the Terminal Complement Complex (TCC) RUO ELISA kit from Svar Life Science AB (Malmö, Sweden) according to the protocol provided by the manufacturer. Briefly, plasma samples were diluted in assay diluent, and 100 µl diluted sample was transferred to microtiter wells pre-coated with anti-TCC monoclonal antibody and incubated at room temperature for 60 min. After washing to remove unbound material, HRP-labeled monoclonal secondary antibody was added to detect the TCC bound to the well. After incubation for 30 min, the wells were washed again, and 3,3′, 5,5′ tetramethylbenzidine dihydrochloride substrate was added. The reaction was stopped after 30 min using stop solution provided with the kit and the absorbance was measured at 450 nm and 650 nm. After subtracting the 650 nm values from the 450 nm values, the amount of TCC was determined from the standard curve (supplied in the kit).

### Complement assays

Since GT103 requires an intact immune system to inhibit tumor growth, we analyzed baseline serum complement activity to assure overall competence of the complement cascade in patients. Soluble C5b-9 concentrations, a measure of complement activation and a potential PD marker were assessed in plasma to determine if the change caused by GT103 during treatment was associated with outcomes.

Blood was collected in tubes with a clot activator for isolation of serum. The serum was isolated, aliquoted, and frozen at −80 °C. For the complement assay, serum samples were thawed and placed on ice. Baseline complement activity was assessed using a standard hemolysis assay employing sheep erythrocytes coated with rabbit anti-sheep IgM antibodies. The assay protocol and assay-specific reagents were obtained from Complement Technology Inc. (Tyler, TX). Briefly, serum samples were initially diluted 100-fold in gelatin veronal buffer (GVB) containing physiological concentrations of $Ca^{++}$ and $Mg^{++}$. Further, serial dilutions from 1:150 to 1:500 were incubated with sheep erythrocytes. After a 60 min incubation at 37 °C, cells and membranes were pelleted by centrifugation and the absorbance of the supernatant fraction at 450 nm was determined. Negative and positive controls were obtained from erythrocytes incubated in GVB or water, respectively.

### IHC for complement regulatory proteins

IHC was performed using the Ultra Discovery (Roche) automated staining platform in conjunction with the Roche HRP Detection System, a biotin-free, hapten (HQ)-anti-hapten-based antibody conjugate system for the detection of tissue-bound primary antibodies. The kit includes peroxidase-blocking reagent, post-primary IgG linker reagent (HQ-labeled secondary antibody), and anti-HQ-HRP reagent to localize HQ-labeled secondary antibodies. CFH was detected with muGT103, a murinized version of the human GT103 antibody bearing the identical CDR regions, diluted 1:60,000 with Discovery Ab Diluent (catalog #760-108). CD55 was detected with rabbit polyclonal anti-CD55 from MilliporeSigma (catalog #HPA002190) diluted 1:100 using the same diluent. Tissue sections were treated for epitope retrieval with Roche cell conditioning solution CC1 (Roche catalog #950-124) for 56 min, then incubated with muGT103 or anti-CD55 for 60 min at 37 °C. Mouse (for muGT103) or rabbit (for anti-CD55) IgG were used as negative controls. After binding of mouse or rabbit primary antibodies, Roche anti-mouse HQ (catalog #760-4814) or anti-rabbit HQ (catalog #760-4815) antibodies, respectively, were applied and incubated for 12 min, followed by 12 min incubation with anti-HQ HRP (catalog #760-4820) for antigen detection. The IHC reactions were visualized with DAB chromogen and counterstained with hematoxylin.

IHC slides were evaluated by a pathologist in a blinded manner according to a semi-quantitative immunoreactivity scoring system: Each slide was scored based on the staining intensity from 0 (no staining), 1 (weak), 2 (moderate), to 3 (strong) and the percentages of positive tumor with staining of each intensity (1 to 100%) was used to compute the H score, where H = intensity score × percentage score, creating a composite score from 0 to 300.

### Reporting summary

Further information on research design is available in the Nature Portfolio Reporting Summary linked to this article.

## Data availability

The study protocol is available in the Supplementary Information. Given patient privacy, clinical data cannot be made publicly available. The deidentified dataset is provided as a Source Data file. The remaining data are available within the Article, Supplementary Information or Source Data file. Source data are provided with this paper.

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

## Acknowledgements

The authors acknowledge the contribution of data safety monitoring board members Dr. Sirish Gadgeel, Martin Edelman, and Mary Redman and are grateful to the participating patients and their families. Grid Therapeutics contributed to trial conception, design, supplied the drug and funded the clinical trial. Research Immunohistochemistry Service BioRepository & Precision Pathology Center (BRPC) performed all of the IHC for this study.

## Author contributions

J.M.C. was the study primary investigator. J.M.C., E.F.P., and J.E.H. contributed to the study design and development concept. G.R.S., H.M., T.E.S., N.R., J.C., G.S., S.A. were clinical investigators of the study. J.M.C. drafted the manuscript. J.E.H. and L.G. performed statistical analysis. S.B. perform pharmacologic analysis. M.C., R.S., E.B.G., E.F.P., A.N., H.L. and Y.W.H. were responsible for laboratory studies and made intellectual contributions to the manuscript. All authors contributed to the acquisition, analysis or interpretation of data, and performed critical revision of the manuscript for intellectual content.

## Competing interests

This research was supported by Grid Therapeutics. Drs. Patz, Campa, and Gottlin have affiliation with both Grid Therapeutics and Duke University. JMC reports trial funding from Bristol-Myers Squibb, Genentech, Spectrum, Adaptimmune, AbbVie, Moderna, GlaxoSmithKline, Array, AstraZeneca, Grid Therapeutics, Abel Zeta, Pfizer; advisory/consulting from AstraZeneca, Merck, Pfizer, Spectrum, Genentech, Novartis, Turning Point, G1 Therapeutics, Vivacitas, Omega, Amgen, Corbus, Sanofi; speaking and travel from Merck and Amgen; DSMB from BioThera and G1 Therapeutics. GS reports speaking for AstraZeneca, consulting for Daiichi Sankyo, board member for Florida chapter of ASCO, and stock Options and consulting compensation for Onc.AI. HM reports advisory board for AstraZeneca, Genentech, and Daiichi Sankyo; and research funding from AstraZeneca. TS reports clinical trial funding from AstraZeneca, Seagen, Mirati Therapeutics, Genentech/Roche, Nuvalent, Inc.; consulting from Takeda, G1 Therapeutics, Spectrum Pharmaceuticals, Gilead Sciences, AstraZeneca, Coherus Biosciences, Blueprint Medicines, Boehringer Ingelheim, Pfizer, Abbvie; travel funding from Pfizer; DSMB participation from Genentech. NR reports advisory compensation from BMS, Merck, Jazz, Genentech, Daiichi, Regeneron, ABBVIE, research funding Merck, Regeneron, and speaking from Jazz. JC reports Scientific Advisor from Actimed, Enzychem, Gen Sci, Pfizer, Tensegrity; DSMB Member of BioAtla, G1 Therapeutics; PI/Institutional Research Funding for AstraZeneca, Helsinn, Pfizer, NCI/NCTN; Publications Committee of Amgen, Frensenius-Kabi, G1, Pfizer. GS reports advisory board from EMD Serono, BMS, Merck, Seattle Genetics, Astellas, Janssen, Bicycle Therapeutics, Pfizer, Gilead, Scholar Rock, Eli Lilly, Loxo Oncology, Vial, Aktis, Daiichi-Sankyo; Consultant/Scientific Advisory Board (SAB)/trial steering committee: Syapse, Merck, Servier, Syncorp, Ellipses; Research Support to institution from EMD Serono, Jazz Therapeutics, Bayer, Sumitomo Pharma, Blue Earth Diagnostics; Speaker from Seagen, Gilead, Natera, Exelixis, Janssen, Astellas, Bayer, Aveo, Pfizer, Merck; Data safety monitoring committee (honorarium) from Mereo; Employment: Spouse employed by Myriad, Exact Sciences; and Travel: BMS, Astellas. SB receives support from the National Institutes of Health, the Childhood Arthritis and Rheumatology Research Alliance

(CARRA), serves on an NIH DSMB, and consults for UCB (Morrisville, NC, USA), CARRA, and Rutgers University. Stephen Balevic is a member of the American College of Rheumatology, the Childhood Arthritis and Rheumatology Research Alliance (CARRA), Alpha Omega Alpha, the American Society of Clinical Investigators, and the Society for Pediatric Research, and served as the Assistant Scientific Director for CARRAAN reports funding from Genentech, Genmab, MedImmune/ AstraZeneca, Seattle Genetics; consulting from Sanofi, Promega; SAB from Leap Therapeutics; NCI Chair, Core Correlatives Sciences Committee (NCTN-CCSC); and patents of No 925592 and No. 62/337,633. MC is a founder of Grid Therapeutics, LLC and is on the advisory board. EBG is a founder of Grid Therapeutics, LLC. SA reports being member of scientific advisory boards for Cellepus Therapeutics (and Co-founder), Leap Therapeutics, Guardian Bio, Immutep, Shoreline Therapeutics, Tubulis, Xilis, Glympse Bio, Achilles Therapeutics, Memgen, RAPT Biotherapeutics. EFP is a founder, CEO and on the BOD of Grid Therapeutics, LLC. All other authors report no competing interests.
