## [Transparent Peer Review file · Nature Communications]

Complement Factor H Targeting Antibody GT103 in Refractory Non-Small Cell Lung Cancer: a Phase 1b Dose Escalation Trial

Corresponding Author: Dr Jeffrey Clarke

Version 0:

Reviewer comments:

Reviewer #1

(Remarks to the Author)

Reviewer Summary:

This manuscript describes the design and results of a phase 1, dose escalation trial to assess the safety of an IgG3 monoclonal antibody, GT103, as treatment for advanced stage, refractory non-small cell lung cancer. The trial's primary aims were to determine the maximum tolerated dose and assess safety of this treatment. The trial used a simple 3+3 design as the basis for dose escalation decisions.

The study design and statistical analysis approach are appropriate for this small dose-finding study. The dose escalation procedure and safety, pharmacokinetic, and treatment response data are presented clearly overall. Certain aspects of the design and analysis need clarification, though. Additionally, the technical details about the assays conducted to measure GT103 and complement are quite cumbersome to read for non-specialists; they are not needed to understand the objectives, design, nor results of the study and would be better moved to Supplemental Material or omitted altogether.

Abstract:

- The Methods paragraph calls the 10mg/kg every 2 weeks and 15mg/kg every 3 weeks dose levels "dose expansion", but 5 patients were treated at each of these levels according to the 3+3 design and the protocol states that the dose expansion part of the trial was removed from the trial design in an amendment. These levels should not be called dose expansion cohorts but rather as part of the dose escalation scheme.

Methods section:

- "Following the dose escalation phase, and based on safety and PK results, an amendment was implemented (November 2022) to incorporate two additional dose levels at 15 mg/kg every 3 weeks and 10 mg/kg every 2 weeks."

Similar as previous comment - the protocol describes the additional dose levels as being part of the dose escalation phase / MTD finding and so they should be described as such.

- "Predefined adverse events (AEs) that were considered dose-limiting toxicity (DLT) if occurring during the first cycle of treatment and deemed to be possibly, probably, or definitely related to study treatment as determined by the investigator." It is unclear what the authors are trying to communicate here.

Assessments section:

- The dose level of 10mg/kg every 2 weeks should not be called level 4 since its dosing frequency differs from the true level 4 dose of 10mg/kg every 3 weeks. The former should be labeled something else (e.g. level 4a as stated in the protocol).

Pharmacokinetic analysis section:

- Last sentence: It is unclear what Stage 1a and 1b mean. Are these the two dose levels involving 10 mg/kg, called Levels 4 and 4a in the protocol? Needs clarification.

Treatment section:

- How many patients discontinued treatment? It is stated that 90% discontinued due to progression, but the denominator is not stated.

- The 95% confidence interval for the median PFS is stated to be 6.0 - 6.1 weeks, which seems impossibly narrow for a cohort of only 31 patients. Please verify this is accurate.

Pharmacokinetics section:

- “The PK of intravenous GT103 after single administration in adult patients with refractory, advanced stage NSCLC was best characterized using a 3-compartment population PK model....”

Clarify how this model was determined to be the best choice among the candidate models.

Exploratory Pharmacodynamic and Immunohistochemical Correlative Analysis:

- “A statistically significant association between baseline levels of sC5b-9 or day 15 change and PFS was not observed.” How was this association tested? Clarification should be added to the Statistics section for this analysis.

- “Of note, higher levels of CD55 were associated with shorter overall survival (HR 1.022, 95% CI 1.003-1.043, p=0.026).”

The analysis method used (presumably a Cox model?) should be described in the Statistics section, along with all variables tested for association and procedures used to check model assumptions.

Supplemental Figure 3 says that these results are for PFS, not OS. The language in this paragraph and Supplemental Figure 3 needs to be reconciled.

Discussion section:

- The first paragraph would fit better in the Introduction section as relevant background material.

- Paragraph 6: “An exploratory analysis of molecular characteristics of the patients’ tumors observed higher PFS for tumors with a KRAS mutation, but these findings did not reach statistical significance due to the small number of patients.”

“But these findings did not reach statistical significance due to the small number of patients” is pure speculation and should be removed - an equally plausible explanation is that the higher observed PFS for KRAS mutated tumors occurred purely by chance, especially given that multiple mutations were analyzed. Neither the results sections nor tables/figures include the statistical test of association of KRAS with PFS referenced here.

- Paragraph 9, last sentence: As noted previously, the trial removed its dose expansion phase and so reference to it should be omitted from here.

Tables and Figures:

- Please add percentages to Table 2 to clarify the prevalence of each type of AE.

- For all figures, the tick mark labels, axis labels, and other text are too small. These need to be enlarged for legibility.

- The curves for dose level 0.3mg/kg appear to be missing from Supplemental Figures 1a and 1b.

General Issue:

- Several acronyms are not defined at their first appearance and should be clarified at first presentation; some of note are CI, OS, HUS, IV, PK, CBC, MR, IHC, V.

- Numerous grammatical errors and typos exist throughout the paper that need correction.

Reviewer #2

(Remarks to the Author)

The manuscript “First-in-Human Clinical Trial Targeting Complement Factor H with the IgG3 Antibody GT103 in Refractory Advanced Stage Non-Small Cell Lung Cancer” by Clarke and co-workers presents intriguing and significant findings, particularly due to its first-in-human nature. Exploring GT103’s safety and preliminary efficacy provides valuable insights into novel therapeutic avenues for patients with advanced NSCLC. I believe that this work deserves publication in a decent journal, however, there are some points that need a bit more clarification.

Major points:

Introduction: For the readers familiar with general principles of the complement system but not familiar with prior authors’ work on FH autoantibodies, it might be confusing that intravenous administration of antibodies targeting the main soluble complement inhibitor present at approx. 0.5 mg/ml does not result in the formation of immunocomplexes with circulating FH but affects tumor cells. To understand the concept, the Authors should bring the information that this antibody most probably binds the cryptic epitope present on FH bound to the tumor cell membrane (as explained in ref. 9). Also, information that this is a function-blocking antibody that disables C3b binding would be helpful, next to the information that some of the lung cancer cells produce endogenous FH (PMID: 15342420), and that tumor cells may actively increase local FH concentration not only by endogenous production but also by hijacking a fully active FH from their microenvironment (PMID: 23760402 and 15342420). All of this information may persuade the readers why the impairment of FH is so important for the host’s response to lung cancer cells. Some of the abovementioned info is present in line 71, but in my opinion, this is too brief and deserves an underlining.

Line 64: the sentence about FH overexpression of CFH in tumor cell membrane is misleading, this is a soluble complement inhibitor that can be rebound to the cell surface.

Methods/Results: Due to the potential importance of these findings, and in light of recent publications on statistical analysis plans (<https://www.bmj.com/content/376/bmj-2021-068177>) and reporting standards (<https://www.bmj.com/content/383/bmj-2023-076387>) for early phase dose-finding clinical trials, I suggest that the authors refer to these guidelines. Preparing the appropriate checklists would ensure comprehensive and transparent reporting of their study methods and findings, thus enhancing the manuscript’s rigour and clarity.

Regarding the statistical analysis plan, I would additionally appreciate an explanation of the choice of the 3+3 dose-escalation design. This traditional method has several limitations, and there are suggestions to use more advanced approaches, such as model-based designs like the continual reassessment method or model-assisted designs such as a modified toxicity probability interval design (https://ascopubs.org/doi/10.1200/EDBK_319783). These alternatives can provide more accurate estimations of the maximum tolerated dose and improve the efficiency of dose-finding studies. Lines 261-262: This is the description of a hemolytic assay for the classical complement pathway settings, whereas FH is the inhibitor of the alternative complement pathway. Diagnostic laboratories report separated assessments of AP 50 and CP 50 values, and the hemolytic assay with Ig-sensitized sheep erythrocytes is a surrogate for the latter. Should not the Authors also perform a hemolytic assay on rabbit erythrocytes that spontaneously activate AP? Especially, when it is not known, which pathway is mainly responsible for the elimination of lung cancer cells in vivo. Please comment on that.

Minor comments and suggestions:

- Exclusion Criteria in Figure 1: The term "brain metastasis" appears unclear, as the mere presence of metastasis did not exclude patients. Similarly, the phrase "tumor size" needs clarification—does it refer to a specific threshold or condition that was part of the exclusion criteria? This also applies to "prohibited medications" - a more detailed explanation of what constitutes "prohibited medication" would be helpful.

- PD-L1 status and immunotherapy: I noted that a significant proportion of the patients had low ($\leq 1\%$) or untested PD-L1 expression levels, yet received immunotherapy. In many countries, such patients might not typically receive immunotherapy due to potential lack of efficacy. It would be valuable to understand how these patients were treated and the rationale for including such a population in the study, especially given the potential for GT103 to synergize with immunotherapy, as suggested by the ongoing phase II study.

- DCR interpretation in the Discussion: the Authors report a DCR of 29%, but I suggest a more tempered interpretation. In this heavily pretreated, yet fit population with an ECOG status of 0-1, many patients were likely "slow progressors", which could explain the lack of progression according to RECIST criteria over a relatively short observation period. Additionally, stable disease (SD), which is the best outcome observed in the trial, can still involve an increase in tumor size, which should be considered when interpreting DCR.

The nomenclature of the C5b-9 (TCC) assay is not consistent in the text, e.g. line 219 it is sC5-9, then it is C5b-9. But more importantly, was the plasma with EDTA as an anticoagulant used for this assay? Was the blood for obtaining the plasma samples processed fast and were the resulting plasma samples immediately aliquoted and kept at -80 C until the experiment? This is critical for the reliability of the assay and omitting TCC formation in the test tube, so the Authors should include a clear statement on that.

I believe addressing these points will strengthen the manuscript and provide the scientific community with a clearer understanding of the study.

Reviewer #3

(Remarks to the Author)

The clinical trial described the safety profile of GT103, a first-in-class, fully human, IgG3 monoclonal antibody targeting complement factor H. A subset of patients could benefit from this novel treatment. Major revision was recommended.

1. There are some mistakes in the writing. In the sentence "Sixty-one percent of the patients had adenocarcinoma histology and 32% had prior history of brain metastasis at time of enrollment" on page 6, sixty-one percent should be modified as 61%. The authors should delete the "?" in "Prior radiation for metastatic brain lesion?, n (%)" in In table 1. The authors should check the article carefully and correct the writing mistakes.

2. Please discuss the potential patients that might benefit from GT103 and reasons in the discussion.

Reviewer #4

(Remarks to the Author)

Many thanks for the opportunity to review this very good work of this great academic group, with a very attractive IND with a novel and attractive mechanism of action

This is a well-written manuscript on a first-in-human study of GT103, a fully human-derived monoclonal IgG3 antibody, first ever of its class in Oncology therapeutics, that has demonstrated anticancer activity in vitro and in vivo, particularly in NSCLC through tumor specific complement activation, antibody dependent cellular phagocytosis, and complement dependent cytotoxicity, as well as promotion of anti-tumor immunity through different mechanisms. The authors are to be congratulated on this, as well as in the good conduction of the study.

Having said that, there is a major conceptual issue regarding the identification of the optimal dose and schedule of the IND for the following clinical studies, the primary objective of a Ph1 study, which is of genuine importance when the effects of the drug are not clinically "visible" because of lack of apparent toxicity and antitumor activity, as it is the case for GT103. In this study the maximum tolerated dose was not reached because of its excellent tolerance, and, then, the RPh2D of GT103 was

chosen to be an intermediate dose level (10 mg q3w) where only 6 patients were treated:

- What was the rationale of selecting this dose/schedule as the optimal biological one? PKs? PDs? A combination of clinical and pharmacological results?
- Why the two additional higher dose levels (15 mg/kg q3w, and 10 mg/kg q2w) were prematurely closed when only 5 patients were treated in each arm? These might have provided with additional valuable information to further fine tune the recommendation of an optimal dose/schedule for next studies with the agent.
- FDA's Project Optimus recommendations include the possibility of a small randomized part within a Ph1 study of these characteristics to further compare 2 or 3 dose levels to better assess the recommendation of an optimal regimen for Ph2 studies. We wonder why this was not done here, as we think it might have provided with additional light on such a key question for the successful posterior clinical development of an IND

A second major aspect to consider is the fact that, although the IND showed an excellent tolerance profile, there are no objective data in the results of the study supporting that the drug does what is supposed to be doing in terms of antitumor effect:

- No objective tumor remissions as per RECIST
- Disease control rate (DCR) is a non-validated endpoint for antitumor activity
- Time-dependent endpoints like overall survival (OS) or progression-free survival (PFS) also lack validity when there is not a control arm, in the context of a ph1 study

And DCR, OS and PFS are the main parameters used in the manuscript to speak about the antitumor activity of the drug, which, even with it, is not remarkable. In addition, it is unknown if the drug achieved significant exposure levels in the patients that might correlate with antitumor activity in preclinical testing, and, finally, the PD results (sC5b-9) are irregular and do not show a pattern of dose- or exposure-dependent effect that might help to build up a mechanistic path of the IND.

Other less relevant comments on this interesting manuscript are:

- How did the authors deal with accumulative toxicity that might be, in theory, limiting further administration of the same dose of the IND that might have occurred after the DLT window (i.e., late toxicities)? As the escalation model was a non-bayesian one, this might be of relevance depending on the toxicity profile of the drug. This is not described in the text or in the specific table, where, also, the % of TRAEs are not shown.
- Regarding one of the two DLTs that were observed, the one that consisted in grade 3 acute kidney failure (prerenal) as a consequence of grade 2 IND-related colitis should be more properly labeled as grade 3 colitis, instead.
- We also recommend to include a PK table/graphic, as it is usually easier to understand by the readership

Reviewer #5

(Remarks to the Author)

Reviewer #6

(Remarks to the Author)

Version 1:

Reviewer comments:

Reviewer #1

(Remarks to the Author)

I thank the authors for their replies to my comments. The revised manuscript is much improved and my remarks have been addressed. I have no further comments or recommended edits.

Reviewer #2

(Remarks to the Author)

I am satisfied with the Authors' reply. They have addressed all my comments and improved the manuscript accordingly.

Reviewer #4

(Remarks to the Author)

I have gone through the response from the co-authors, and they have done a great effort with my questions, which I appreciate and recognize.

Having said that, in the end, they agree that they could not optimize yet the recommended phase 2 dose of the IND, on one hand, and, on the other, no formal testing or results of preliminary antitumor activity are seen. And these are weak points for a Phase 1 study.

Reviewer #6

(Remarks to the Author)

Response to Reviewers

General revisions: We have moved the Methods section to after the Discussion to conform with the Nature style, and also put the references in the Nature style. We have checked all the spelling, grammar, punctuation, and abbreviations. Since a lot of minor corrections of this type were made these were accepted in the Tracked Changes version for readability. Changes of substance were left marked in Tracked Changes. The abstract was shortened to conform to journal requirements. We have moved some Discussion material to the Introduction, and the Introduction was expanded to make the background of the project more understandable. A new Supplementary Methods document was created as suggested, and the figures and tables have been improved.

Reviewer #1 (Remarks to the Author): with expertise in clinical trial study design, biostatistics

Reviewer Summary:

This manuscript describes the design and results of a phase 1, dose escalation trial to assess the safety of an IgG3 monoclonal antibody, GT103, as treatment for advanced stage, refractory non-small cell lung cancer. The trial's primary aims were to determine the maximum tolerated dose and assess safety of this treatment. The trial used a simple 3+3 design as the basis for dose escalation decisions.

The study design and statistical analysis approach are appropriate for this small dose-finding study. The dose escalation procedure and safety, pharmacokinetic, and treatment response data are presented clearly overall. Certain aspects of the design and analysis need clarification, though. Additionally, the technical details about the assays conducted to measure GT103 and complement are quite cumbersome to read for non-specialists; they are not needed to understand the objectives, design, nor results of the study and would be better moved to Supplemental Material or omitted altogether.

We have clarified the design and analysis throughout in response to the reviewers' comments. As suggested, we have moved the technical details about the assays conducted to measure GT103 and complement to Supplementary Methods.

Abstract:

- The Methods paragraph calls the 10mg/kg every 2 weeks and 15mg/kg every 3 weeks dose levels “dose expansion”, but 5 patients were treated at each of these levels according to the 3+3 design and the protocol states that the dose expansion part of the trial was removed from the trial design in an amendment. These levels should not be called dose expansion cohorts but rather as part of the dose escalation scheme.

We agree with this comment and have edited the line appropriately, substituting “dose escalation” for “dose expansion”. (page 7)

Methods section:

- “Following the dose escalation phase, and based on safety and PK results, an amendment was implemented (November 2022) to incorporate two additional dose levels at 15 mg/kg every 3 weeks and 10 mg/kg every 2 weeks.”

Similar as previous comment - the protocol describes the additional dose levels as being part of the dose escalation phase / MTD finding and so they should be described as such.

As suggested the Methods section has been edited to clarify that the additional dose levels were part of the dose escalation. (page 7)

- “Predefined adverse events (AEs) that were considered dose-limiting toxicity (DLT) if occurring during

the first cycle of treatment and deemed to be possibly, probably, or definitely related to study treatment as determined by the investigator.”

It is unclear what the authors are trying to communicate here.

The above sentence was rewritten more clearly as: “AEs as defined in the trial protocol were considered DLTs if they occurred during the first cycle of treatment and were deemed to be possibly, probably, or definitely related to study treatment as determined by the investigator.” (page 7)

We refer to the trial protocol in this new sentence, and the NCT protocol number is given in the manuscript. To elaborate here, DLT as defined in the protocol must be at least possibly related to GT103 treatment per the investigator. The authors have included protocol text below regarding the complete definition of the AEs considered to be DLT for reference:

- Hematologic toxicity: Any grade 4 neutropenia, thrombocytopenia or anemia or grade ≥ 3 neutropenia or thrombocytopenia lasting over 7 days
- Any grade 3 thrombocytopenia associated with bleeding
- Febrile neutropenia (grade 3/4)
- Nausea/Vomiting or Diarrhea \geq grade 3 **and** lasting ≥ 3 days despite adequate supportive measures
- Grade ≥ 3 ALT or AST elevation
- Other non-hematologic toxicity \geq grade 3 excluding alopecia, anorexia, fatigue, lab abnormalities deemed not clinically significant by the Investigator
- Rare, idiosyncratic reactions to the study drug (grade 3/4).
- Fatigue (lasting longer than 7 days) and hypertension will be considered as DLT only if they ≥ 3 or are considered unmanageable.
- Treatment delay of > 7 days in cycle 1 related to toxicity. (Note: If there are treatment delays, the study calendar will move).
- Any treatment-related death or treatment-related hospitalization
- Grade 4 anaphylaxis, hypotension, hypoxia or acute respiratory distress”

Assessments section:

- The dose level of 10mg/kg every 2 weeks should not be called level 4 since its dosing frequency differs from the true level 4 dose of 10mg/kg every 3 weeks. The former should be labeled something else (e.g. level 4a as stated in the protocol).

As suggested, edits have been made in the manuscript. We have done away with the stage 1a/b and level 4/4a terminology.

Pharmacokinetic analysis section:

- Last sentence: It is unclear what Stage 1a and 1b mean. Are these the two dose levels involving 10 mg/kg, called Levels 4 and 4a in the protocol? Needs clarification.

We are no longer using this terminology. GT103 PK assay was performed and analyzed in two batches (before and after the amendment): In the first batch were dosing levels 0.3 to 10 mg/kg every 3 weeks, and in the second batch were dosing levels 10mg/kg every 2 weeks and 15 mg/kg every 3 weeks. We have clarified this in the paper.

Treatment section:

- How many patients discontinued treatment? It is stated that 90% discontinued due to progression, but the denominator is not stated.

All 31 patients were off treatment at the time the current analysis was done. The breakdown is below and this information has been added to text of the manuscript.

Reason for off treatment, n (%)

Disease progression	28 (90.3)
New primary developed	1 (3.2)
Decline in performance and lack of benefit from treatment	1 (3.2)
Withdrawal/refusal after beginning protocol therapy	1 (3.2)

- The 95% confidence interval for the median PFS is stated to be 6.0 - 6.1 weeks, which seems impossibly narrow for a cohort of only 31 patients. Please verify this is accurate.

We appreciate the reviewer's comment but the biostatisticians have verified the results.

Pharmacokinetics section:

- "The PK of intravenous GT103 after single administration in adult patients with refractory, advanced stage NSCLC was best characterized using a 3-compartment population PK model...."
- Clarify how this model was determined to be the best choice among the candidate models.

When deciding on a final structural model, we rely on several model diagnostic criteria including the objective function value, plausibility of parameter estimates, residual (unexplained) error, and model diagnostic plots.

Compared to a 1-CPT model (OFV 4150), the 2-CPT (OFV 4116) and 3-CPT (OFV 4086) models had improved performance. The 2-CPT model was statistically better than the 1-CPT model with a p-value of 4.18E-08; and the 3-CPT model was statistically better than the 2-CPT model with a p-value of 4.44E-07. Residual error was 23.1% for the 1 CPT model, 20.6% for the 2 CPT model, and 19.3% for the 3 CPT model. Parameter precision was overall good across all models, with the exception of CL2 (31.3%) and V2 (77.1%, CI containing 0) for the 2-CPT model. Accordingly, the 3-CPT model with the lowest OFV and generally good parameter precision was selected as the final model.

Exploratory Pharmacodynamic and Immunohistochemical Correlative Analysis:

- "A statistically significant association between baseline levels of sC5b-9 or day 15 change and PFS was not observed."

How was this association tested? Clarification should be added to the Statistics section for this analysis.

A univariate Cox proportional hazard model was used to assess the association between baseline levels of sC5b-9 or day 15 change and PFS. The statistical methods for all biomarkers were added to the Methods section. (page 9)

- "Of note, higher levels of CD55 were associated with shorter overall survival (HR 1.022, 95% CI 1.003-1.043, p=0.026)."

The analysis method used (presumably a Cox model?) should be described in the Statistics section, along with all variables tested for association and procedures used to check model assumptions.

This analysis method has been added to the methods section. (page 9) The CD55 analysis included PFS and has been clarified in the manuscript.

Supplemental Figure 3 says that these results are for PFS, not OS. The language in this paragraph and Supplemental Figure 3 needs to be reconciled.

The correlation of CFH and CD55 are with PFS. This as been reconciled in the text. (page 4)

Discussion section:

- The first paragraph would fit better in the Introduction section as relevant background material.

We agree and appreciate the reviewers comment. Accordingly, we have rewritten the Introduction and the Discussion: The Introduction focuses on relevant background material (and is expanded in scope from the previous version) and the Discussion focuses more on the results.

- Paragraph 6: “An exploratory analysis of molecular characteristics of the patients’ tumors observed higher PFS for tumors with a KRAS mutation, but these findings did not reach statistical significance due to the small number of patients.”

"But these findings did not reach statistical significance due to the small number of patients" is pure speculation and should be removed - an equally plausible explanation is that the higher observed PFS for KRAS mutated tumors occurred purely be chance, especially given that multiple mutations were analyzed. Neither the results sections nor tables/figures include the statistical test of association of KRAS with PFS referenced here.

We agree with the reviewer, and this has been edited to delete “due to the small number of patients”. (page 5)

- Paragraph 9, last sentence: As noted previously, the trial removed its dose expansion phase and so reference to it should be omitted from here.

Agree, and this has been edited.

Tables and Figures:

- Please add percentages to Table 2 to clarify the prevalence of each type of AE.

As suggested, percentages have been added.

- For all figures, the tick mark labels, axis labels, and other text are too small. These need to be enlarged for legibility.

As suggested, these improvements have been made.

- The curves for dose level 0.3mg/kg appear to be missing from Supplemental Figures 1a and 1b.

The dose level is presented in the updated figures.

General Issue:

- Several acronyms are not defined at their first appearance and should be clarified at first presentation; some of note are CI, OS, HUS, IV, PK, CBC, MR, IHC, V.

Acronyms have now been defined at first appearance.

- Numerous grammatical errors and typos exist throughout the paper that need correction.

We apologize, and these have been corrected throughout the manuscript.

Reviewer #2 (Remarks to the Author): with expertise in complement biology and lung cancer

The manuscript "First-in-Human Clinical Trial Targeting Complement Factor H with the IgG3 Antibody GT103 in Refractory Advanced Stage Non-Small Cell Lung Cancer" by Clarke and co-workers presents intriguing and significant findings, particularly due to its first-in-human nature. Exploring GT103's safety and preliminary efficacy provides valuable insights into novel therapeutic avenues for patients with advanced NSCLC. I believe that this work deserves publication in a decent journal, however, there are some points that need a bit more clarification.

Major points:

Introduction: For the readers familiar with general principles of the complement system but not familiar with prior authors' work on FH autoantibodies, it might be confusing that intravenous administration of antibodies targeting the main soluble complement inhibitor present at approx. 0.5 mg/ml does not result in the formation of immunocomplexes with circulating FH but affects tumor cells. To understand the concept, the Authors should bring the information that this antibody most probably binds the cryptic epitope present on FH bound to the tumor cell membrane (as explained in ref. 9).

Also, information that this is a function-blocking antibody that disables C3b binding would be helpful, next to the information that some of the lung cancer cells produce endogenous FH (PMID: 15342420), and that tumor cells may actively increase local FH concentration not only by endogenous production but also by hijacking a fully active FH from their microenvironment (PMID: 23760402 and 15342420). All of this information may persuade the readers why the impairment of FH is so important for the host's response to lung cancer cells. Some of the abovementioned info is present in line 71, but in my opinion, this is too brief and deserves an underlining.

We appreciate the reviewer's comments on tumor specificity as a function of a cryptic epitope and on the mechanism of action of the anti-FH antibody. We have now put more background material in the Introduction regarding both of these topics. Regarding tumor specificity, *in vitro* evidence suggests this antibody binds a cryptic epitope in CFH (Campa et al., *Cancer Imm. Res.*, 2015; Bushey et al., *Cell Reports*, 2016), binds tumor tissue but not normal tissue (Bushey et al., *MCT*, 2023), and does not bind soluble CFH (Bushey et al., *MCT*, 2023). We therefore postulate that the tumor specificity observed *in vivo* is the result of epitope exposure when CFH is bound to the tumor cell membrane. We have also made a FH knockout cell line and seen that the GT103 antibody that inhibits tumor growth of the wild type cell line does not inhibit tumor growth of the knockout cell line (Saxena et al., *J. Immunol.*, 2024). This suggests that the antibody binding to endogenous, tumor cell-expressed CFH is required for its activity.

Regarding mechanism, when we first discovered the autoantibody and epitope mapped it to a C3b binding site in CFH, our initial hypothesis was that the antibody must block this site, allowing C3b to be deposited and thus promoting the alternative pathway of complement activation (Campa et al., *CIR*, 2015). By carrying out tumor cell CDC assays *in vitro* in complete serum vs. factor B- or C1q-depleted serum, we found that this hypothesis was wrong: *In vitro*, the GT103 antibody requires C1q but not factor B to promote tumor cell lysis and therefore, GT103 works by the classical pathway (Bushey, et al. *MCT*, 2023). In other words, *in vitro*, GT103 is not a blocking antibody, but an antibody that binds an exposed cryptic epitope in CFH and then interacts with C1q via its Fc domain to trigger the classical pathway. *In vivo*, GT103 may work by more than one mechanism to bring about CDC.

Line 64: the sentence about FH overexpression of CFH in tumor cell membrane is misleading, this is a soluble complement inhibitor that can be rebound to the cell surface.

CFH can either be overexpressed by tumor cells or bound from the fluid phase to the cell surface (as shown in, for example, PMID:15342420 – Ajona et al, 2004). But the wording has now been edited in

accordance with the reviewer's comment. The sentence now reads "In addition, high levels of CFH in the tumor cell membrane are associated with a poor prognosis". (page 2)

Methods/Results: Due to the potential importance of these findings, and in light of recent publications on statistical analysis plans (<https://www.bmj.com/content/376/bmj-2021-068177>) and reporting standards (<https://www.bmj.com/content/383/bmj-2023-076387>) for early phase dose-finding clinical trials, I suggest that the authors refer to these guidelines. Preparing the appropriate checklists would ensure comprehensive and transparent reporting of their study methods and findings, thus enhancing the manuscript's rigour and clarity.

We appreciate the above references and have reviewed the checklists accordingly for our publication.

Regarding the statistical analysis plan, I would additionally appreciate an explanation of the choice of the 3+3 dose-escalation design. This traditional method has several limitations, and there are suggestions to use more advanced approaches, such as model-based designs like the continual reassessment method or model-assisted designs such as a modified toxicity probability interval design (https://ascopubs.org/doi/10.1200/EDBK_319783). These alternatives can provide more accurate estimations of the maximum tolerated dose and improve the efficiency of dose-finding studies.

The authors agree that modern dose escalation schemes, particularly, Bayesian approaches, can confer advantages for novel therapeutic development. Indeed, such designs would be valid for dose escalation of GT103 and were discussed at the time of development approximately 6 years ago. However, ultimately the 3+3 design has relative advantages of providing increased number of patients at each dose level for toxicity assessment, PK analysis, predictive biomarker development, predictable dose levels for drug manufacturing, and efficacy signal seeking purposes. Furthermore, given the high degree of novelty of the target, immune effects, and IgG3 construct, a rapid dose escalation design may have posed risk for encountering less frequent adverse events.

Lines 261-262: This is the description of a hemolytic assay for the classical complement pathway settings, whereas FH is the inhibitor of the alternative complement pathway. Diagnostic laboratories report separated assessments of AP 50 and CP 50 values, and the hemolytic assay with Ig-sensitized sheep erythrocytes in a surrogate for the latter. Should not the Authors also perform a hemolytic assay on rabbit erythrocytes that spontaneously activate AP? Especially, when it is not known, which pathway is mainly responsible for the elimination of lung cancer cells in vivo. Please comment on that.

The AP is initiated by the spontaneous hydrolysis of C3. Factor B, factor D, and properdin are then required for full AP activity. However, deficiencies in these factors are very rare. Since the pathway beyond C3 cleavage into C3a and C3b is identical in the CP and AP, we chose to use CP as a surrogate for total complement activity. This analysis was only performed at baseline to assure patients had a competent complement system.

Minor comments and suggestions:

- Exclusion Criteria in Figure 1: The term "brain metastasis" appears unclear, as the mere presence of metastasis did not exclude patients. Similarly, the phrase "tumor size" needs clarification—does it refer to a specific threshold or condition that was part of the exclusion criteria? This also applies to "prohibited medications" - a more detailed explanation of what constitutes "prohibited medication" would be helpful.

"Tumor size" is referring to one patient who did not have measurable disease for eligibility. We have clarified this in the manuscript. One patient also started methotrexate which was contraindicated medication. (page 10)

The full eligibility criteria are below regarding the brain metastasis in the protocol. Patients may have brain metastases but be ineligible due to a number of potential clinical scenarios.

Symptomatic brain or leptomeningeal metastases, including patients who continue to require glucocorticoids and/or antiseizure therapy for brain or leptomeningeal metastases.

1. **Treated**, asymptomatic metastases are permitted provided the patient has completed radiation at least 2 weeks prior to day 1 and has been off steroids for at least 2 weeks prior to day 1 of study drug.
2. **Untreated** brain metastases are permitted if asymptomatic, do not require immediate treatment, steroids and/or antiseizure medications, and are ≤ 1 cm.

- PD-L1 status and immunotherapy: I noted that a significant proportion of the patients had low ($\leq 1\%$) or untested PD-L1 expression levels, yet received immunotherapy. In many countries, such patients might not typically receive immunotherapy due to potential lack of efficacy. It would be valuable to understand how these patients were treated and the rationale for including such a population in the study, especially given the potential for GT103 to synergize with immunotherapy, as suggested by the ongoing phase II study.

Prior treatment with immunotherapy was required for eligibility for the trial. Since this is a first-in-man trial, primarily designed for safety and tolerability we wanted to include as many patients as possible. PD-L1 negative patients can still derive benefit from immune checkpoint therapies, and the effect of GT103 is unknown. Additionally, PD-L1 IHC is not known to be a biomarker of activity for GT103. The hypothesis of the study is that GT103 would provide clinical benefit and anti-tumor effect in immune refractory tumors regardless of PD-L1 status.

- DCR interpretation in the Discussion: the Authors report a DCR of 29%, but I suggest a more tempered interpretation. In this heavily pretreated, yet fit population with an ECOG status of 0-1, many patients were likely "slow progressors", which could explain the lack of progression according to RECIST criteria over a relatively short observation period. Additionally, stable disease (SD), which is the best outcome observed in the trial, can still involve an increase in tumor size, which should be considered when interpreting DCR.

We appreciate and agree with the reviewers comment, and this is an important issue. We have now added additional caveat to the findings in the Discussion regarding "slow progressors". (page 5)

The nomenclature of the C5b-9 (TCC) assay is not consistent in the text, e.g. line 219 it is sC5-9, then it is C5b-9.

We have fixed this in the manuscript.

But more importantly, was the plasma with EDTA as an anticoagulant used for this assay? Was the blood for obtaining the plasma samples processed fast and were the resulting plasma samples immediately aliquoted and kept at -80 C until the experiment? This is critical for the reliability of the assay and omitting TCC formation in the test tube, so the Authors should include a clear statement on that.

Thank you, we have now elaborated on the collection methods. Blood was collected in EDTA tubes for plasma and in tubes with a clot activator for serum. The plasma and serum were isolated, aliquoted, and frozen at -80 °C. For the assays, plasma or serum samples were thawed and placed on ice. We have added this information for the sC5b-9 (TCC) and GT103 PK assays (where plasma was used) and the total complement assay (where serum was used) to the new Supplementary Methods section.

I believe addressing these points will strengthen the manuscript and provide the scientific community with a clearer understanding of the study.

Reviewer #3 (Remarks to the Author): with expertise in lung cancer, therapy

The clinical trial described the safety profile of GT103, a first-in-class, fully human, IgG3 monoclonal antibody targeting complement factor H. A subset of patients could benefit from this novel treatment. Major revision was recommended.

1. There are some mistakes in the writing. In the sentence “Sixty-one percent of the patients had adenocarcinoma histology and 32% had prior history of brain metastasis at time of enrollment” on page 6, sixty-one percent should be modified as 61%. The authors should delete the “?” in “Prior radiation for metastatic brain lesion?, n (%)” in In table 1. The authors should check the article carefully and correct the writing mistakes.

As suggested, writing mistakes have been corrected in the manuscript.

2. Please discuss the potential patients that might benefit from GT103 and reasons in the discussion.

We appreciate the reviewer’s comment and have now added a section in the Discussion that addresses the potential patients who might benefit from GT103. We have stated in the discussion that patients with metastatic NSCLC may benefit more in earlier line of treatment and benefit will be increased with addition of immune checkpoint therapy. (page 6)

Reviewer #4 (Remarks to the Author): with expertise in lung cancer, therapy

Many thanks for the opportunity to review this very good work of this great academic group, with a very attractive IND with a novel and attractive mechanism of action

This is a well-written manuscript on a first-in-human study of GT103, a fully human-derived monoclonal IgG3 antibody, first ever of its class in Oncology therapeutics, that has demonstrated anticancer activity in vitro and in vivo, particularly in NSCLC through tumor specific complement activation, antibody dependent cellular phagocytosis, and complement dependent cytotoxicity, as well as promotion of anti-tumor immunity through different mechanisms. The authors are to be congratulated on this, as well as in the good conduction of the study.

Having said that, there is a major conceptual issue regarding the identification of the optimal dose and schedule of the IND for the following clinical studies, the primary objective of a Ph1 study, which is of genuine importance when the effects of the drug are not clinically "visible" because of lack of apparent toxicity and antitumor activity, as it is the case for GT103. In this study the maximum tolerated dose was not reached because of its excellent tolerance, and, then, the RPh2D of GT103 was chosen to be an intermediate dose level (10 mg q3w) where only 6 patients were treated:

- What was the rationale of selecting this dose/schedule as the optimal biological one? PKs? PDs? A combination of clinical and pharmacological results?

The reviewer raises an excellent point. Since this is the first-in-man trial, the first IgG3, and no known MTD, is it currently extremely difficult with this number of patients to choose the optimal dose and schedule. We selected the RPh2D based on preclinical studies that suggested 10 mg/kg would be reasonable for biologic activity, and a dosing schedule that was matched with other immune therapies so

that a patient friendly combination study could be performed. Once we have more data, we plan on modeling the PK and PD results to optimize dosing parameters. We have added this information in the Discussion. (page 6)

- Why the two additional higher dose levels (15 mg/kg q3w, and 10 mg/kg q2w) were prematurely closed when only 5 patients were treated in each arm? These might have provided with additional valuable information to further fine tune the recommendation of an optimal dose/schedule for next studies with the agent.

We have included in the manuscript an explanation of the early closure:

“Of note, the protocol was closed to enrollment prior to completing the 15 mg/kg every 3 weeks and 10 mg/kg every 2 weeks dose levels (5 patients treated at each level). This decision was jointly made by the sponsor and PI based on competing studies and projected accrual timeline, and not for toxicity reasons.” (page 7)

- FDA's Project Optimus recommendations include the possibility of a small randomized part within a Ph1 study of these characteristics to further compare 2 or 3 dose levels to better assess the recommendation of an optimal regimen for Ph2 studies. We wonder why this was not done here, as we think it might have provided with additional light on such a key question for the successful posterior clinical development of an IND

The authors agree regarding the importance of dose optimization and the Project Optimus initiative. The current study was designed and started enrollment well before Project Optimus was begun. Especially given the MTD was not reached, future trials with a randomized approach should be used to assess activity and safety between dose levels for GT103.

A second major aspect to consider is the fact that, although the IND showed an excellent tolerance profile, there are no objective data in the results of the study supporting that the drug does what is supposed to be doing in terms of antitumor effect:

- No objective tumor remissions as per RECIST
- Disease control rate (DCR) is a non-validated endpoint for antitumor activity
- Time-dependent endpoints like overall survival (OS) or progression-free survival (PFS) also lack validity when there is not a control arm, in the context of a ph1 study

And DCR, OS and PFS are the main parameters used in the manuscript to speak about the antitumor activity of the drug, which, even with it, is not remarkable. In addition, it is unknown if the drug achieved significant exposure levels in the patients that might correlate with antitumor activity in preclinical testing, and, finally, the PD results (sC5b-9) are irregular and do not show a pattern of dose- or exposure-dependent effect that might help to build up a mechanistic path of the IND.

Thanks so much for your comment. These are very interesting, important and complicated issues, and highlight the difficult in interpreting these other parameters and efficacy in phase 1 data. The **trial was designed primarily for safety and tolerability**. Since GT013 has a very different mechanism of action than traditional chemotherapy, not only causing tumor cell death but a robust immune response, the most appropriate metric to measure response and efficacy is uncertain. In fact, prior studies have shown that early response in advanced stage NSCLC does not correlate with overall survival (Birchard et al. PMID: 19117348). Since lung tumors have a widely variable cellular composition (Christensen, et al. PMID: 20533438), conventional imaging does not accurately reflect the pathological response.

Pre and post treatment biopsies unfortunately were not feasible for this small phase 1 study due to the difficulty often encountered with performing biopsies in lung cancer patients. We have measured sC5b-9

as a candidate PD biomarker to gain insight into on-target effects of GT103 as discussed in the manuscript. Clearly future studies that confirm PD biomarkers and incorporation of pre- and post-biopsy specimens in a subset of patients is needed to confirm drug efficacy. Ultimately, a randomized study with a standard of care control arm is needed to establish clinical efficacy.

Other less relevant comments on this interesting manuscript are:

- How did the authors deal with accumulative toxicity that might be, in theory, limiting further administration of the same dose of the IND that might have occurred after the DLT window (i.e., late toxicities)? As the escalation model was a non-bayesian one, this might be of relevance depending on the toxicity profile of the drug. This is not described in the text or in the specific table, where, also, the % of TRAEs are not shown.

Thank you for the comment. Patients were continued to be followed for toxicity outside of the DLT period. The toxicity profile seen cumulatively shows the drug to still be well tolerated with low rate of grade 3 AE (and no grade 4 or 5 toxicity). Future studies (including the ongoing phase 2 combination study) will add to the knowledge of toxicity profile with this agent including late side effects..

- Regarding one of the two DLTs that were observed, the one that consisted in grade 3 acute kidney failure (prerenal) as a consequence of grade 2 IND-related colitis should be more properly labeled as grade 3 colitis, instead.

The CTCAE grading of colitis was felt to be most consistent with grade 2 AE by the investigator based on the definition. The grade 3 definition of colitis does not include hospitalization or organ injury. The additional acute kidney failure grading accurately characterizes the toxicity profile experienced by the subject.

- We also recommend to include a PK table/graphic, as it is usually easier to understand by the readership. We thank the reviewer for this suggestion. We have included a graph of the GT103 concentration over time profile, as well as a table that characterizes the exposures observed throughout the study in the supplemental materials. As the reviewer noted, this is usually the highest yield data and easiest to understand by readers.

Reviewer #5 (Remarks to the Author):

Reviewer #6 (Remarks to the Author):
